# Perioperative outcomes and causes of postpartum hemorrhage in patients undergoing cesarean delivery in Thailand: A comprehensive retrospective study

**Patchareya Nivatpumin**[1☺]*, **Jitsupa Nithi-Uthai**[1☺], **Tripop Lertbunnaphong**[2‡], **Nattapon Sukcharoen**[1‡], **Thanyarat Soponsiripakdee**[1‡], **Pawana Yonphan**[1‡]

**1** Faculty of Medicine Siriraj Hospital, Department of Anesthesiology, Mahidol University, Bangkok, Thailand,
**2** Faculty of Medicine Siriraj Hospital, Department of Obstetrics and Gynecology, Mahidol University, Bangkok, Thailand

☺ These authors contributed equally to this work.
‡ These authors also contributed equally to this work
* patchareya.niv@mahidol.ac.th

**Data Availability Statement:** All relevant data are within the manuscript.

## Abstract

### Background

This study aimed to identify the characteristics, causes, perioperative anesthetic, and obstetric outcomes of patients experiencing postpartum hemorrhage (PPH) after cesarean delivery.

### Methods

We retrospectively analyzed patients who underwent cesarean delivery at the largest university hospital in Bangkok, Thailand, during a 5-year period (January 1, 2016–December 31, 2020). PPH was defined as an estimated blood loss (EBL) of $\geq$ 1000 ml within 24 hours postpartum.

### Results

Of 17 187 cesarean deliveries during the study period, 649 patients were included for analysis. The mean EBL was 1774.3 ± 1564.4 ml (range: 1000–26 000 ml). Among the patients, 166 (25.6%) experienced massive PPH (blood loss > 2000 ml). Intraoperative blood transfusions were necessary for 264 patients (40.7%), while 504 individuals (77.7%) needed intraoperative vasopressors. The analysis revealed uterine atony as the leading cause of PPH in 62.7% (n = 407) of the patients, with abnormal placentation following at 29.3% (n = 190). Abnormal placentation was associated with a significantly higher mean EBL of 2345.0 ± 2303.9 ml compared to uterine atony, which had a mean EBL of 1504.0 ± 820.7 ml ($P$ < 0.001). Abnormal placentation also significantly increased the likelihood of blood transfusions and hysterectomies ($P$ < 0.001 for both) and led to more intensive care unit admissions ($P$ = 0.032). The risk of EBL exceeding 2000 ml was markedly higher in patients with abnormal placentation (odds ratio [OR] 5.12, 95% confidence interval [CI] 3.45–7.57, $P$ <

**Funding:** The author(s) received no specific funding for this work.

**Competing interests:** The authors have declared that no competing interests exist.

0.001) and in cases involving trauma to the internal organs (OR 2.33, 95% CI 1.16–4.71, *P* = 0.018) than in patients with uterine atony. The study documented three instances of perioperative cardiac arrest, one of which was fatal.

## Conclusions

These findings highlight the importance of comprehensive perioperative management strategies, including the ready availability of adequate blood and blood products, particularly in scenarios predisposed to significant hemorrhage.

## Trial registration

Clinical trial registration: Clinicaltrial.gov registration number NCT04833556 (April 6, 2021).

## Introduction

Postpartum hemorrhage (PPH) is the leading cause of maternal cardiac arrest and death in Sub-Saharan Africa and parts of Asia [1–4]. Countries with lower sociodemographic metrics typically exhibit a higher prevalence of maternal mortality [5]. In Thailand, maternal death rates ranged from approximately 20.0 to 40.5 per 100 000 deliveries between 1990 and 2015 [5]. Hemorrhage has been reported to account for 20%–30% of maternal deaths in Thailand [2]. PPH is defined as bleeding of 500 ml or more following vaginal delivery and 1000 ml or more after cesarean delivery [4]. The general prevalence of PPH, irrespective of the delivery method, lies between 2.9% and 3.2% [6,7], whereas in cesarean deliveries, the rate is approximately 0.4% to 5.1% [6,8,9]. The reported worldwide prevalence can reach as high as 10% [10].

The literature offers comprehensive insights into various etiological factors behind PPH across delivery modes [4,6]. The major causes of PPH can be categorized into four groups: uterine atony, placental causes, traumatic injuries to internal organs or the birth passage, and coagulopathy [4]. Uterine atony is widely recognized as the primary cause, accounting for an estimated 70% to 80% of cases [4,6]. Profuse bleeding resulting from PPH can lead to severe complications, including disseminated intravascular coagulation, the need for extensive blood transfusions, acute renal failure, multi-organ failure, prolonged hospital stays, and the necessity for intensive care unit admission [4,6,11–14].

Thailand falls within the mid-range of the sociodemographic index [5], and Siriraj Hospital, located in the capital city, is the largest university hospital in the country. As the primary referral hub for many hospitals in Bangkok and nearby provinces, the hospital holds a pivotal position in the region's healthcare network. The hospital manages approximately 7500 to 8000 deliveries annually.

No studies in Thailand have reported the clinical characteristics, perioperative outcomes, and causes of PPH following cesarean delivery. This research at Siriraj Hospital aimed to bridge this knowledge gap. We identified the perioperative outcomes, including anesthetic and obstetrical results, and determined the distribution of PPH cases by cause.

## Methods

We conducted a retrospective analysis. Before this research began, its protocol was authorized by the Siriraj Institutional Review Board of the Faculty of Medicine, Siriraj Hospital, Mahidol University, Bangkok, Thailand (protocol number 005/2564(IRB4), approval number Si 161/

2021). The study was also registered at www.clinicaltrials.gov (NCT04833556). The study adhered to the Strengthening the Reporting of Observational Studies in Epidemiology (STROBE) guidelines.

## Study population

We accessed the electronic medical records of patients who underwent cesarean deliveries at Siriraj Hospital between January 1, 2016, and December 31, 2020. Records were identified using the International Classification of Disease (ICD-10) code O72.1, labeled "other immediate postpartum hemorrhage." Only records with this ICD-10 code were included in the analysis. We excluded patients with a gestational age of less than 24 weeks and those with incomplete anesthetic records. Data on the volume of blood loss were extracted from anesthetic records, recovery room logs, and patient charts for the first 24 postoperative hours. Data collection for the retrospective analysis of patient records from January 1, 2016, to December 31, 2020, commenced on April 22, 2021.

## Definitions

The study employed the following definitions:

- **Primary postpartum hemorrhage:** Bleeding amounting to 1000 ml or more within 24 hours following a cesarean delivery [4].

- **Causes of PPH:** These were divided into four main categories (4-T classification) [4].

- **Tone:** Uterine atony, defined as the need for two or more uterotonic agents.

- **Tissue:** Abnormal placentation, defined as the diagnosis of placenta previa, placenta accreta spectrum, or other documented placental bleeding issues.

- **Trauma:** Bleeding due to injuries to the birth passage or internal organs.

- **Thrombin:** Bleeding resulting from coagulopathy or hemostatic problems.

- **Massive hemorrhage:** Bleeding amounting to 2000 ml or more.

- **Hypotension:** A systolic blood pressure below 90 mmHg or a drop of 20% or more from the preanesthetic level. The incidence and severity of hypotension were measured by determining the number of boluses of any vasopressor administered (e.g., ephedrine, noradrenaline, or adrenaline) from anesthetic records.

- **Bradycardia:** Instances when atropine was administered or when the patient's heart rate dropped below 50 beats per minute.

- **Blood transfusion:** The administration of at least one unit of packed red cells to a patient.

- **Neonatal birth asphyxia:** A neonatal Apgar score lower than 7 at 5 minutes after delivery.

- **Transfusion-related acute lung injury:** A new onset of acute lung injury within 6 hours of transfusion, where other causes were ruled out [15].

- **Acute kidney injury:** Diagnosed following the Kidney Disease: Improving Global Outcomes (KDIGO) 2012 clinical practice guidelines for acute kidney injury definition, which are based on increased serum creatinine levels or reduced urine output [16].

## Statistical analysis

The sample size calculation was based on the incidence of perioperative complications from PPH. Previous studies indicated incidences of blood transfusion ranging from 33% to 54.6% [17–19], peripartum hysterectomy at 52.2% [19], and intensive care unit admission rates at 71.4% [13]. The formula used for calculation was $n = Z(1-\alpha)2\, p\, (1-p)/d2$. Considering an estimated incidence of perioperative complications of 30%, a confidence level of 95%, and allowable errors of 0.04, a sample size of more than 505 charts was determined.

All analyses were performed with PASW Statistics, version 18 (SPSS Inc, Chicago, IL, USA). Categorical data are presented as numbers and percentages. Continuous data are shown as either means ± standard deviations or medians (ranging from minimum to maximum value). Dichotomous variables were subjected to chi-square or Fisher's exact tests. Analysis of variance (ANOVA) or the Kruskal–Wallis test was employed for continuous data comparisons. Post-hoc analysis was performed by pairwise comparison with Bonferroni correction and Benjamini-Hochberg method. Regression analysis was utilized to determine the risk of massive hemorrhage, with results presented as crude odds ratios alongside their 95% confidence intervals (CI). A P value below 0.05 was deemed statistically significant. The Wilson method was used to determine a 95% CI for the incidence of PPH.

## Results

A total of 17 187 cesarean deliveries were performed during the 5-year study period. Of these, 669 patients' charts were retrieved with the diagnosis of immediate PPH (coded as O72.1), but only 649 charts met the criteria for analysis (**Fig 1**). The incidence of PPH was 3.8% (95% CI 3.5–4.1).

The patients' demographics and clinical characteristics are outlined in **Table 1**. The mean estimated blood loss was 1774.3 ± 1564.4 ml, ranging from 1000 to 26 000 ml. Intraoperative clinical and anesthetic data are detailed in **Table 2**. The predominant anesthetic technique employed was single-shot spinal anesthesia, which was used in 419 out of 649 cases (64.6%). However, 41 out of the 649 patients (6.3%) who initially received regional anesthesia needed conversion to general anesthesia. Among those who underwent general anesthesia, 4 patients (0.6%) had difficult airways, and 7 patients (1.1%) experienced intraoperative desaturation.

**Table 3** compares perioperative outcomes segmented by the primary causes of PPH. Abnormal placentation led to the highest mean blood loss, measured at 2345.0 ± 2303.9 ml. The internal organ trauma documented in our findings encompassed tears in the uterine artery, uterine venous plexus, lower segment of the uterus, cervix, placental bed, board ligament, salpinx, and mesosalpinx. Ruptures of ovarian/endometriotic cysts were also noted, as well as damage to other organs, such as the urinary bladder and colon. A comparison of perioperative outcomes based on three of the four principal causes of PPH (uterine atony, abnormal placentation, and trauma to internal organs) is depicted in **Fig 2**.

The logistic regression analysis comparing the three groups is presented in **Table 4**. The odds ratio for massive bleeding ($\geq$ 2000 ml) was higher when comparing abnormal placentation and trauma to the internal organs against uterine atony.

Neonatal outcomes for patients with PPH are detailed in **Table 5**. Of the cases, 35 (4.8%) experienced birth asphyxia, and 13 (1.7%) resulted in neonatal death. No significant association was found between antepartum hemorrhage and neonatal birth asphyxia (P 0.147).

**Table 6** presents the overall postoperative outcomes and prognoses. None of the patients received interventional radiology. Three patients experienced cardiac arrest during the perioperative period, resulting in an incidence rate of 0.46% (95% CI 0.16–1.35). The first patient was diagnosed with amniotic fluid embolism and experienced massive bleeding. The second

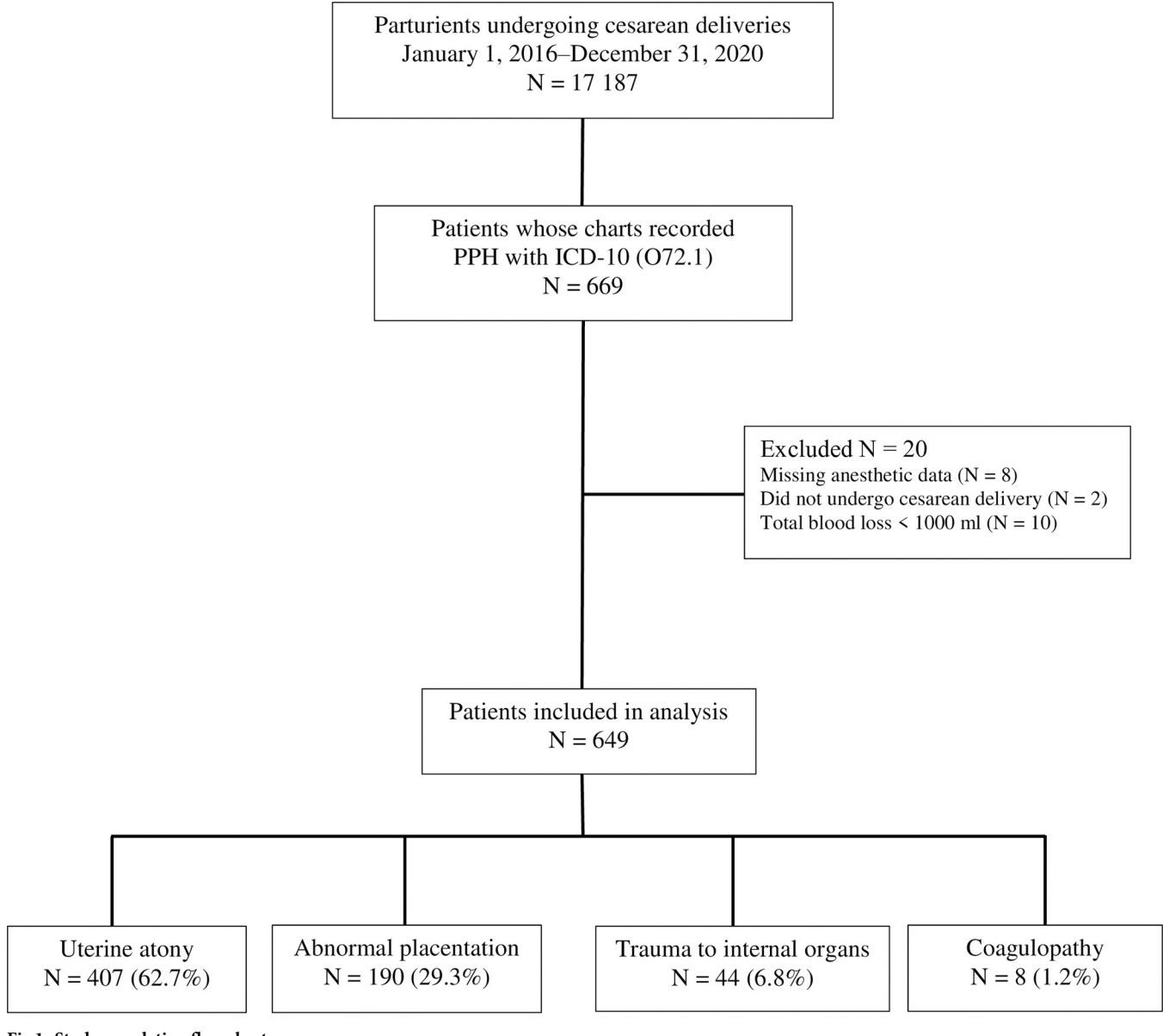

**Fig 1. Study population flow chart.**

patient was diagnosed with placenta percreta, requiring extensive transfusion due to massive bleeding, and suffered an accidental air embolism originating from the peripheral intravenous line. The third patient also had placenta percreta and underwent an emergency cesarean delivery. She suffered from massive bleeding (approximately 26 liters) and unfortunately passed away on the day of surgery.

## Discussion

During the 5 years from 2016 to 2020, our study found that the rate of PPH in patients undergoing cesarean delivery was 3.8%. The two most common causes of PPH at our institute were uterine atony, accounting for 62.7% of cases, and abnormal placentation, accounting for 29.3% of cases. Among the analyzed patients, 166 out of 649 (25.6%) experienced massive PPH

**Table 1. Demographic and clinical characteristics (n = 649).**

| Parameters | Value |
|---|---|
| Age (years) | 33.8 ± 5.6 (17–48) |
| Weight (kilograms) | 73.6 ± 13.9 (45–128) |
| Height (centimeters) | 158.5 ± 6.2 (130–175) |
| BMI (kilograms per square meter) | 29.2 ± 4.9 (19–48.9) |
| Primigravida | 291 (44.8) |
| Gestational age (week) | 37.1 ± 2.7 (25–42) |
| ASA classification | |
| • II | 575 (88.6) |
| • III | 72 (11.1) |
| • IV | 2 (0.3) |
| Elective cesarean delivery (n = 236) | |
| • Previous cesarean delivery | 103 (15.9) |
| • Breech presentation | 26 (4.0) |
| • Fetal transverse/oblique lie | 10 (1.5) |
| • Maternal request | 16 (2.5) |
| • Contracted pelvis | 3 (0.5) |
| • Abnormal placentation | 9 (1.4) |
| • Other[1] | 69 (10.7) |
| Urgent/emergency cesarean delivery (n = 413) | |
| • Cephalopelvic disproportion | 93 (14.3) |
| • Fetal NICHD category 3 | 51 (7.9) |
| • Umbilical cord prolapses | 2 (0.3) |
| • Hypovolemic shock | 4 (0.6) |
| • Thick meconium-stained amniotic fluid | 7 (1.1) |
| • Ruptured uterus | 4 (0.6) |
| • Abruptio placenta | 2 (0.3) |
| • Prolonged premature ruptured of membrane | 14 (2.2) |
| • Previous cesarean delivery with labor pain | 31 (4.8) |
| • Breech presentation with labor pain | 19 (2.9) |
| • Abnormal placentation with labor pain | 46 (7.1) |
| • Triplet with labor pain | 2 (0.3) |
| • Other[2] | 138 (21.3) |
| Twins | 72 (11.1) |
| Triplets | 2 (0.3) |
| Antepartum hemorrhage | 109 (16.8) |
| Pregnancy-associated problem (n = 188) Gestational hypertension Gestational diabetes Preeclampsia | 20 (3.1) 111 (17.1) 57 (8.8) |

Abbreviations: ASA, American Society of Anesthesiologists; BMI, body mass index; NICHD, National Institute of Child Health and Human Development; SD, standard deviation.

Data are presented as mean ± standard deviation (minimum–maximum value) or number (percentage).

[1] *Other indications for elective cesarean delivery were twin, fetal macrosomia, advanced maternal age, myoma uteri, previous uterine surgery, condyloma accuminata at the vulva, death of fetus in utero with placenta previa, fetal oral hemangioma, maternal mental retardation, and no data recorded.*

[2] *Other indications for urgent/emergency cesarean delivery were antepartum hemorrhage, preeclampsia with severe features, hemolysis elevated liver enzyme with low platelet (HELLP) syndrome, unfavorable cervix, vaginal septum with labor pain, myoma previa with labor pain, twin pregnancy with labor pain, previous uterine surgery with labor pain, chorioamnionitis, and no data recorded.*

**Table 2. Anesthetic techniques and intraoperative details (N = 649).**

| Parameters | Value |
|---|---|
| Choice of anesthesia | |
| • General anesthesia | 164 (25.3) |
| • Spinal anesthesia | 419 (64.6) |
| • Epidural anesthesia | 10 (1.5) |
| • Combined spinal epidural anesthesia | 5 (0.8) |
| • General anesthesia with epidural anesthesia | 10 (1.5) |
| • Regional anesthesia conversion to general anesthesia | 41 (6.3) |
| Estimated blood loss (ml) | 1774.3 ± 1564.4<br>1300 (1000–26 000) |
| Number of patients with blood loss $\geq$ 2,000 ml | 166 (25.6) |
| Number of patients with intraoperative hypotension | 501 (77.2) |
| Episode of hypotension (n = 501) | 5.09 ± 5.6 |
| Number of patients administered any vasopressor | 504 (77.7) |
| Total ephedrine (mg) (n = 392) | 22.9 ± 14.56 |
| Total norepinephrine (mcg) (n = 376) | 77.4 ± 375.1 |
| Total IV fluid administration (ml) | 2039.7 ± 1055.8<br>1800 (300–10 500) |
| Total colloid (ml) (n = 302) | 383 ± 325.1<br>500 (0–1500) |
| Number of patients received blood transfusions | 264 (40.7) |
| Number of packed red cell units (n = 264) | 2.25 ± 2.5<br>1 (1–23) |
| Number of patients received fresh frozen plasma | 68 (10.5) |
| Number of patients received platelets | 44 (6.8) |
| Number of patients received uterotonic agents | |
| • Oxytocin | 597 (62) |
| • Carbetocin[#] | 55 (3.5) |
| • Methylergometrine | 380 (5.8) |
| • Salprostone | 127 (19.6) |
| • Misoprostol | 10 (1.5) |

Abbreviations: EBL, estimated blood loss; FFP, fresh frozen plasma; GA, general anesthesia; IV, intravenous; mcg, microgram; mg, milligram; ml; milliliters; PRC, packed red blood cells; RA, regional anesthesia.

[#]*Three patients received both oxytocin and carbetocin.*

Data are presented as mean ± standard deviation, median (minimum–maximum value), or number (percentage).

(blood loss $\geq$ 2000 ml). The overall blood transfusion rate was 40.7%, with 264 out of 649 patients needing transfusions.

The incidence of PPH following cesarean delivery at our institute was comparable to a previous population-based study conducted in Norway, which reported rates ranging from 1.9% to 4.7% [9]. However, the definition of PPH used in that study differed from ours, as they defined PPH as blood loss exceeding 1500 ml, not 1000 ml. On the other hand, our incidence rate is higher than the overall incidence of PPH reported in a large nationwide study conducted in the United States, which found rates of 2.9% across all delivery routes. The same study also identified cesarean section itself as a risk factor for atonic PPH necessitating transfusion [6]. Cesarean delivery is frequently performed following the failure of labor induction. This failed induction can lead to uterine muscle fatigue and an increased need for uterotonic agents, consequently heightening the risk of PPH [9,20].

Ashwal et al demonstrated that urgent cesarean delivery was a risk factor for PPH in patients undergoing cesarean delivery [21]. Such patients often face antenatal complications

**Table 3. Comparison of perioperative outcomes by main causes of postpartum hemorrhage (n = 649).**

| | Uterine atony N = 407 | Abnormal placentation N = 190 | Trauma to internal organs N = 44 | Coagulopathy N = 8 | P |
|---|---|---|---|---|---|
| Antepartum hemorrhage n = 109 | 21 (5.2) | 86 (45.3) | 2 (4.5) | 0 (0) | < 0.001* a, d |
| Blood loss volume (ml) | 1,504.0 ± 820.7 1,200 [1000–1,500] (1,000–7860) | 2,345.0 ± 2,303.9 1800 [1200–2500] (1000–26 000) | 1,943.6 ± 2,219.9 1500 [1000–2000] (1000–15 500) | 1,037 ± 74.4 1000 [1000–1075] (1000–1200) | < 0.001# a, e, f |
| Massive bleeding (≥ 2000 ml) n = 166 | 62 (15.2) | 91 (47.9) | 13 (29.5) | 0 (0) | < 0.001* a, b, d |
| Intraoperative blood transfusion n = 264 | 107 (26.3) | 136 (71.6) | 16 (36.4) | 5 (62.5) | < 0.001* a, c, d |
| Choice of anesthesia: general anesthesia[1] | 81 (19.7) | 117 (61.6) | 15 (34.1) | 2 (25.0) | < 0.001* a, d |
| Intraoperative hypotension n = 501 | 328 (80.6) | 136 (71.6) | 32 (72.7) | 5 (62.5) | 0.056 |
| Intraoperative vasopressor used n = 504 | 334 (82.1) | 132 (69.5) | 33 (75.0) | 5 (62.5) | 0.004* a |
| Hysterectomy within 24 hours n = 106 | 30 (7.4) | 69 (36.3) | 7 (15.9) | 0 (0) | < 0.001* a, b, d |
| Reoperation within 24 hours n = 32 | 22 (5.4) | 6 (3.2) | 4 (9.1) | 0 (0) | 0.285 |
| Postoperative intensive care unit admission n = 44 | 19 (4.7) | 20 (10.5) | 5 (11.4) | 0 (0) | 0.032 * a |
| Postoperative blood transfusion n = 208 | 116 (28.5) | 74 (38.9) | 13 (29.5) | 5 (62.5) | 0.018 * a, c |
| Intrauterine balloon n = 41 | 18 (4.4) | 23 (12.1) | 0 (0) | 0 (0) | 0.003* a |
| B-lynch suture n = 11 | 6 (1.5) | 3 (1.6) | 1 (2.3) | 1 (12.5) | 0.187 |
| Uterine artery ligation n = 33 | 15 (3.7) | 12 (6.3) | 6 (13.6) | 0 (0) | 0.031 * b |
| Neonatal birth asphyxia[2] n = 35 | 17 (4.2) | 10 (5.3) | 6 (13.6) | 2 (25.0) | 0.011 * b, c, d |
| Hospital length of stay (days) | 6.9 ± 8.8 5 [4–7] (2–124) | 8.7 ± 8.3 6 [4.8–9] (2–67) | 7.4 ± 8.3 5 [4–8.8] (1–54) | 9.6 ± 7.2 8 [4.3–11.5] (4–26) | < 0.001# a |

Data are presented as number (percentage), mean ± standard deviation, median [interquartile range], or (minimum–maximum value).

*Chi-square or Fisher's exact tests test

#ANOVA or Kruskall-Wallis test.

Post-hoc analysis using pairwise comparison with Bonferroni correction and Benjamini-Hochberg method

a compared between uterine atony and abnormal placentation group.

b compared between uterine atony and trauma to internal organ group.

c compared between uterine atony and coagulopathy group.

d compared between abnormal placentation and trauma to internal organ group.

e compared between abnormal placentation and coagulopathy group.

f compared between trauma to internal organs and coagulopathy group.

[1] "General anesthesia" included patients who (1) received general anesthesia only, (2) received combined regional and general anesthesia, or (3) failed regional anesthesia and were converted to general anesthesia.

[2] "Neonatal birth asphyxia" was defined as a neonatal Apgar score of less than 7 at 5 minutes after delivery.

such as placental abruption or placenta previa, increasing the likelihood of PPH. Our data indicated that nearly one-fifth (16.8%) of our PPH patients had antepartum hemorrhage. Another study showed that emergency cesarean delivery was associated with a higher risk of

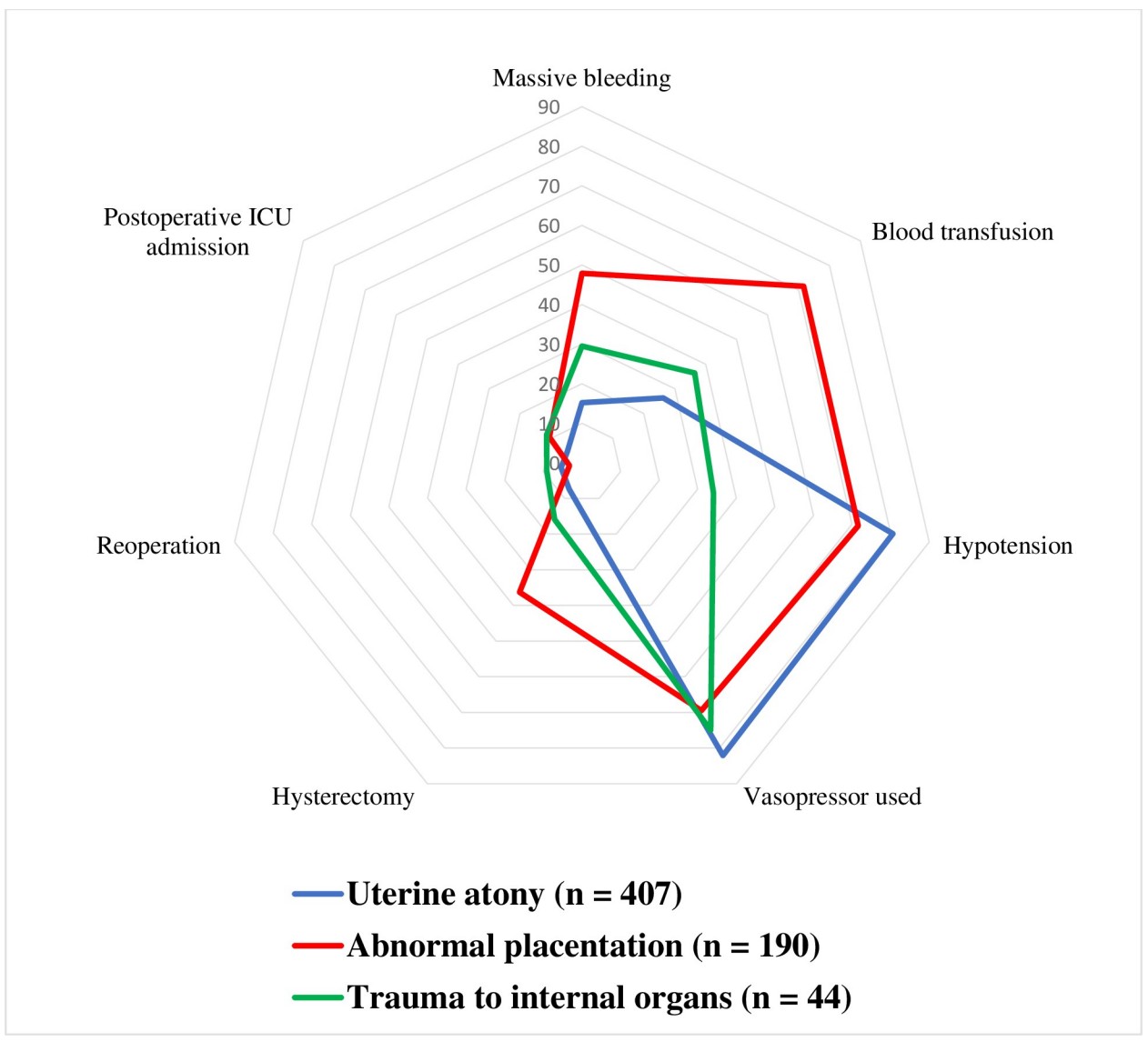

**Fig 2. Perioperative complications by cause: Uterine atony, abnormal placentation, and trauma to internal organs causes (n = 641).**

severe PPH (blood loss exceeding 1500 ml), with an odds ratio of 3.61 [22]. This finding aligns with our data, which revealed that most of our patients (63.6%) underwent urgent or emergency cesarean deliveries.

Uterine atony is the predominant cause of PPH across all delivery methods, accounting for 70% to 80% of cases [4,6]. In our study, however, 62.7% of PPH cases were due to uterine

**Table 4. Odds ratio for massive bleeding ($\geq$ 2000 ml) by causes of postpartum hemorrhage versus uterine atony.**

| Cause | Crude odds ratio (95% confidence interval) | P |
|---|---|---|
| Uterine atony | 1 | |
| Abnormal placentation | 5.12 (3.45–7.57) | < 0.001 |
| Trauma to internal organs | 2.33 (1.16–4.71) | 0.018 |

**Table 5. Neonatal outcomes (n = 725).**

| Parameters | Value |
|---|---|
| Neonatal birth weight (gram) | 2899 ± 708.4 |
| Uterine incision to delivery time (min) | 3.85 ± 3.9 |
| Apgar score at 1 min | 7.55 ± 2.1 |
| Apgar score at 5 min | 8.9 ± 1.7 |
| Birth asphyxia[1] n (%) | 35 (4.8) |
| Neonatal death[2] n (%) | 13 (1.7) |

Data are presented as mean ± standard deviation (minimum–maximum value) or as number (percentage).

[1] *"Birth asphyxia" was defined as a neonatal Apgar score of less than 7 at 5 minutes after delivery.*

[2] *"Neonatal death" included death of fetus in utero (n = 2).*

atony, while 29.3% were attributed to abnormal placentation. This incidence of PPH resulting from abnormal placentation at our institute was notably higher than the approximately 20% reported in another study [6]. This difference can be ascribed to our study's focus: we exclusively analyzed patients who underwent cesarean deliveries. At our hospital, conditions such as placenta previa or placenta accreta typically lead to a recommendation for cesarean delivery. Another contributing factor to the elevated proportion of patients with abnormal placentation is our hospital's stature as a large tertiary care center with advanced facilities. As such, we handle many complicated obstetric cases, including those diagnosed with placenta previa and the placenta accreta spectrum [23].

Globally, cesarean delivery rates have been on the rise [24]. Previous data from our center revealed a substantial cesarean delivery rate of 48.9% in 2017 [25]. Consequently, patients are more likely to develop conditions within the placenta accreta spectrum. The occurrence of this condition is known to increase with the number of prior cesarean deliveries [11]. Past research from our center found that nearly half of the patients (54 out of 113, or 47.8%) with placenta accreta spectrum had undergone previous uterine surgeries, including cesarean deliveries [23].

Our study found a statistically significant increase in the volume of bleeding among patients diagnosed with abnormal placentation compared to those suffering from uterine atony. Notably, 71.6% of patients diagnosed with abnormal placentation and PPH needed at least one unit of packed red blood cell transfusion. A comprehensive systematic review and meta-analysis

**Table 6. Prognosis and maternal outcomes following postpartum hemorrhage (n = 649).**

| Parameters | N (%) |
|---|---|
| Postoperative hypotension | 93 (14.3) |
| Postoperative maternal hemorrhage | 98 (15.1) |
| Postoperative anemia (Hct < 33%) | 361 (55.6) |
| Postoperative blood transfusion | 161 (24.8) |
| Acute kidney injury | 1 (0.2) |
| Transfusion-related acute lung injury | 5 (0.8) |
| Cardiac arrest | 3 (0.5) |
| Postoperative ICU admission | 44 (6.8) |
| Length of hospital stay (days) | 1–124 |
| Maternal death | 1 (0.2) |

Abbreviations: Hct, hematocrit; ICU, intensive care unit.

Data are presented as number (percentage) or as minimum–maximum value.

reported a blood transfusion rate of 46.9% for patients with the placenta accrete spectrum [19]. However, our institution had a markedly higher incidence of blood transfusion than the meta-analysis. This discrepancy can be linked to our study's criteria, which focused on patients with hemorrhages of 1000 ml or more. Additionally, some individuals diagnosed with abnormal placentation in our study experienced antepartum hemorrhages, leading to preoperative anemia and, subsequently, an increased probability of transfusion.

More concerning is the prevalence of severe morbidities resulting from PPH due to abnormal placentation. Our data showed that patients with abnormal placentation had higher rates of maternal morbidity, including hysterectomy and intensive care unit admissions, than those with other causes of PPH. These findings align with the 2018 obstetric care consensus from the American College of Obstetricians and Gynecologists for the placenta accreta spectrum. The consensus highlights the complications associated with the placenta accreta spectrum: massive blood transfusions, transfusion-related acute lung injury, congestive heart failure, acute kidney injury, and multi-organ failure [11]. Moreover, the odds of experiencing massive bleeding were significantly greater for patients with abnormal placentation and trauma-related causes than for those with PPH due to uterine atony. The risks associated with these two primary causes have heightened concerns among anesthesiologists regarding fluid, blood, and blood product preparations during the perioperative period. Additionally, our data revealed a sizable proportion of patients experiencing intraoperative hypotension (77.2%) and receiving vasopressors (77.7%). These findings emphasize the importance of being prepared for intravenous fluid resuscitation and vasopressor administration, as well as providing comprehensive intraoperative anesthetic management for patients with PPH.

In individuals diagnosed prenatally with placenta accreta spectrum, the choice of anesthesia for cesarean delivery depends on various factors, including the patient's comorbidities, the extent of placental invasion, and resource availability [26–28]. Regional anesthesia is typically the preferred technique for cesarean delivery; however, it often leads to maternal hypotension [29–31]. In cases where there is an expectation of massive intraoperative bleeding or planned cesarean hysterectomy, general anesthesia may be more appropriate [32,33]. Although general anesthesia is linked to severe PPH (bleeding $\geq$ 1500 ml) in cesarean deliveries, our data indicated a markedly higher use of general anesthesia in cases of abnormal placentation than in those attributed to uterine atony or trauma-related causes [34]. Factors such as preexisting antepartum bleeding, which can lead to preoperative hypotension, and the expectation of significant hemorrhage from abnormal placentation could prompt anesthesiologists to lean toward general anesthesia for such patients [27,28].

Additionally, placental abnormalities can trigger antepartum hemorrhage and maternal hypotension, which may adversely affect fetal health [35,36]. Our investigation found a higher incidence of antepartum hemorrhage in the abnormal placentation group than in the uterine atony or trauma-related group. Moreover, we observed that 4.8% of neonates experienced birth asphyxia, and 1.7% died. However, our research did not identify a significant relationship between maternal antepartum hemorrhage and neonatal birth asphyxia.

Postpartum hemorrhage is a leading cause of maternal mortality worldwide [2–4]. The Global Burden of Disease study highlighted a substantial decrease in maternal mortality rates in Thailand, dropping from 40.5 per 100 000 deliveries in 1990 to 20.0 per 100 000 deliveries in 2005 [5]. Within these statistics, PPH accounts for 20% to 30% of all causes of maternal mortality [2]. However, despite this progress, there was a case of maternal death among the 649 patients in our study. This particular case was due to a massive hemorrhage from the placenta percreta. Patients who do not have a prenatal diagnosis of placenta accrete spectrum disorders and undergo emergency cesarean delivery are particularly vulnerable to severe hemorrhage. Such situations demand rapid blood and blood product transfusions and precise anesthetic

management. Early identification of hemorrhage requiring massive transfusion is crucial for optimizing patient outcomes.[4,37–39] Collaborative consultations involving obstetricians, anesthesiologists, hematologists, transfusion medicine specialists, and neonatologists should be promptly initiated [11,26–28,37–40].

A strength of our study lies in its comprehensive reporting of patient outcomes in the perioperative period, encompassing preoperative status, intraoperative anesthetic and obstetric management, and postoperative prognosis. Moreover, our analysis of the causes of PPH revealed the major consequences associated with each cause. Nonetheless, there are limitations to our study. The retrospective nature of our study constrains the granularity of data, with details such as the indications for blood transfusion and pretransfusion hemoglobin levels remaining unknown. Anesthesia techniques, intraoperative management, and obstetric interventions varied according to the discretion of the attending anesthesiologists and obstetricians, introducing a degree of clinical variability. Moreover, as the study is based on data from a single tertiary referral center, it may not capture the full spectrum of PPH incidences, outcomes, and causes of cesarean deliveries throughout Thailand, where regional and hospital-level variations are likely. A more expansive national study examining perioperative outcomes from countrywide data on cesarean deliveries is recommended.

## Conclusions

In cesarean delivery cases that resulted in PPH, there was a substantial incidence of massive bleeding and a pronounced need for blood transfusions. Among the causes, abnormal placentation was the leading contributor. Abnormal placentation was also associated with increased maternal mortality, a higher frequency of extensive hemorrhage, elevated rates of blood transfusion, higher rates of hysterectomy, and a greater likelihood of admission to the intensive care unit. Consequently, healthcare personnel must anticipate and be well prepared for perioperative complications, especially in scenarios with elevated risks of intense bleeding.

## Acknowledgments

The authors express gratitude to our hospital director for granting permission to access and distribute patient data. Special thanks go to Mr. Suthiphol Udomphunthurak for his invaluable assistance with statistical analysis, and appreciation is extended to Miss Chusana Rungjindamai for her help with administrative tasks.

## Author Contributions

**Conceptualization:** Patchareya Nivatpumin, Nattapon Sukcharoen.

**Data curation:** Patchareya Nivatpumin, Nattapon Sukcharoen, Thanyarat Soponsiripakdee, Pawana Yonphan.

**Formal analysis:** Patchareya Nivatpumin, Tripop Lertbunnaphong.

**Investigation:** Patchareya Nivatpumin, Tripop Lertbunnaphong, Nattapon Sukcharoen.

**Methodology:** Patchareya Nivatpumin, Jitsupa Nithi-Uthai, Tripop Lertbunnaphong, Nattapon Sukcharoen.

**Project administration:** Patchareya Nivatpumin, Tripop Lertbunnaphong.

**Resources:** Patchareya Nivatpumin, Jitsupa Nithi-Uthai.

**Supervision:** Patchareya Nivatpumin.

**Validation:** Patchareya Nivatpumin, Tripop Lertbunnaphong, Nattapon Sukcharoen.

**Visualization:** Patchareya Nivatpumin.

**Writing – original draft:** Patchareya Nivatpumin, Jitsupa Nithi-Uthai, Tripop Lertbunna-phong, Nattapon Sukcharoen, Thanyarat Soponsiripakdee.

**Writing – review & editing:** Patchareya Nivatpumin, Tripop Lertbunnaphong.

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
