## [Decision Letter · Decision Letter 0]

30 Jan 2024

PONE-D-23-33818Perioperative outcomes and causes of postpartum hemorrhage in patients undergoing cesarean delivery in Thailand: a comprehensive retrospective studyPLOS ONE

Dear Dr. Nivatpumin,

Thank you for submitting your manuscript to PLOS ONE. After careful consideration, we feel that it has merit but does not fully meet PLOS ONE’s publication criteria as it currently stands. Therefore, we invite you to submit a revised version of the manuscript that addresses the points raised during the review process.

We look forward to receiving your revised manuscript.

Kind regards,

Salvatore Andrea Mastrolia, M.D.

Academic Editor

PLOS ONE

Journal Requirements:

Reviewers' comments:

Reviewer's Responses to Questions

**Comments to the Author**

1. Is the manuscript technically sound, and do the data support the conclusions?

Reviewer #1: Yes

2. Has the statistical analysis been performed appropriately and rigorously? 

Reviewer #1: Yes

3. Have the authors made all data underlying the findings in their manuscript fully available?

Reviewer #1: Yes

4. Is the manuscript presented in an intelligible fashion and written in standard English?

Reviewer #1: Yes

5. Review Comments to the Author

Reviewer #1: Very impressive study with an interesting findings,

I would like to highlight some points that need to be modified so that the paper could be more clear for the readers:

1- Could you please write the inclusion and exclusion criteriae for the sample size that you included into your study?

2- What are the limitations of your study?

3- 12 out of your 37 references are up to dat, could you please update some of your references?

4- Reference N. 16 needs to be written in a correct citation style.

6. PLOS authors have the option to publish the peer review history of their article (what does this mean?). If published, this will include your full peer review and any attached files.

Reviewer #1: **Yes: **Mena Abdalla

---

## [Author Response · Author response to Decision Letter 0]

8 Feb 2024

Prof. Emily Chenette

Editor-in-Chief

PLOS ONE

February 7th, 2024

Re: PLOS ONE Manuscript PONE-D-23-33818

Dear Prof. Emily Chenette,

Please find attached the revised version of the original article “Perioperative outcomes and causes of postpartum hemorrhage in patients undergoing cesarean delivery in Thailand: a comprehensive retrospective study” for consideration for publication in PLOS ONE.

Your comments were highly insightful and enabled us to improve the quality of our manuscript markedly. In the following pages are our point-by-point responses to the reviewer's comment. Details and revisions in the text are shown using the red font.

We have incorporated comprehensive details into the manuscript and its associated discussion, following the valuable suggestions provided by the reviewers. Additionally, we have added the inclusion and exclusion criteria in the method part. Furthermore, we have added the limitation of the study in the discussion part. The updated references have also been added (reference numbers #10, 13, 14, and 15), and rewritten all the reference styles to the PLOS ONE format by Endnote X20 for Mac. 

We hope that the revisions made to this article, along with our accompanying response have met the necessary criteria to render our manuscript suitable for publication in PLOS ONE.

We look forward to hearing from you at your earliest convenience.

Yours sincerely,

Patchareya Nivatpumin, M.D.

Associate Professor

Division of Obstetric Anesthesia

Department of Anesthesiology

Faculty of Medicine Siriraj Hospital

Mahidol University

2 Prannok Road, Bangkok, Thailand 10700

Tel: +66 89 666 2187; Fax: +66 2 411 3256

E-mail: patchareya.niv@mahidol.ac.th

Responses to the comments of Reviewers

1. Could you please write the inclusion and exclusion criteriae for the sample size that you included into your study?

Response: We have added the inclusion and exclusion criteria in the method part. “We accessed the electronic medical records of patients who underwent cesarean deliveries at Siriraj Hospital between January 1, 2016, and December 31, 2020. Records were identified using the International Classification of Disease (ICD-10) code O72.1, labeled “other immediate postpartum hemorrhage.” Only records with this ICD-10 code were included in the analysis. We excluded patients with a gestational age of less than 24 weeks and those with incomplete anesthetic records. 

2. What are the limitations of your study?

Response: We have added the limitation of the study in the last paragraph of the discussion part. “Nonetheless, there are limitations to our study. The retrospective nature of our study constrains the granularity of data, with details such as the indications for blood transfusion and pretransfusion hemoglobin levels remaining unknown. Anesthesia techniques, intraoperative management, and obstetric interventions varied according to the discretion of the attending anesthesiologists and obstetricians, introducing a degree of clinical variability. Moreover, as the study is based on data from a single tertiary referral center, it may not capture the full spectrum of PPH incidences, outcomes, and causes of cesarean deliveries throughout Thailand, where regional and hospital-level variations are likely.” A more expansive national study examining perioperative outcomes from countrywide data on cesarean deliveries is recommended.”

3. 12 out of your 37 references are up to date, could you please update some of your references?

Response: We have incorporated the revised reference list, encompassing citations 10, 13, 14, and 15. Notably, the definition of Transfusion-related acute lung injury (TRALI) has undergone modification, with reference 15 now reflecting this change: 'A consensus redefinition of transfusion-related acute lung injury. Transfusion. 2019;59(7):2465-76.' This update has been implemented accordingly. However, it is pertinent to highlight that the definition of acute kidney injury (AKI) remains consistent with the previously published iteration from 2012, as delineated in 'KDIGO clinical practice guidelines for acute kidney injury. Nephron Clin Pract. 2012;120(4):c179-84.' This aspect has been maintained without alteration.

4. Reference N. 16 needs to be written in a correct citation style.

Response: The references have all been reformatted to adhere to the PLOS ONE citation style using EndNote X20.

---

## [Editor Report · Decision Letter 1]

4 Mar 2024

Perioperative outcomes and causes of postpartum hemorrhage in patients undergoing cesarean delivery in Thailand: a comprehensive retrospective study

PONE-D-23-33818R1

Dear Authors,

We’re pleased to inform you that your manuscript has been judged scientifically suitable for publication and will be formally accepted for publication once it meets all outstanding technical requirements.

Kind regards,

Salvatore Andrea Mastrolia, M.D.

Academic Editor

PLOS ONE

---

## [Editor Report · Acceptance letter]

4 Apr 2024

PONE-D-23-33818R1 

PLOS ONE

Dear Dr. Nivatpumin, 

I'm pleased to inform you that your manuscript has been deemed suitable for publication in PLOS ONE. Congratulations! Your manuscript is now being handed over to our production team.

Kind regards, 

on behalf of

Dr. Salvatore Andrea Mastrolia 

Academic Editor

PLOS ONE